# Adolescent Employment, Mental Health, and Suicidal Behavior: A Propensity Score Matching Approach

**DOI:** 10.3390/ijerph17186835

**Published:** 2020-09-18

**Authors:** Hyerine Shin, Kyung hee Kim, Ji-su Kim, Eunkyung Lee

**Affiliations:** 1Department of Nursing, Chung-Ang University, 84 Heukseok-ro Dongjak-Gu, Seoul 06974, Korea; impluos@cau.ac.kr (H.S.); jisu80@cau.ac.kr (J.-s.K.); 2Department of Nursing, Kyung-in Women’s University, 63 Gyeyangsan-ro, Gyesan-dong, Seoul 06974, Korea; eklee@kiwu.ac.kr

**Keywords:** part-time work, mental health, suicidal behavior, adolescents

## Abstract

This study aimed to confirm the relationships between part-time work experience, mental health, and suicidal behavior in adolescents. The impact of part-time work in this population is a controversial topic, perhaps because of the sociocultural background-related inconsistencies in previous results. In this cross-sectional study, which involved a secondary analysis of data from the 11th–13th Korea Youth Risk Behavior Web-Based Surveys, conducted among 800 middle and high schools by the Korean government, we used propensity score matching analysis to minimize the impact of individual backgrounds on the findings concerning the relationships in question. Overall, part-time experience was significantly related to mental health problems and suicidal behavior even after eliminating background differences. Adolescents with part-time work experience had higher overall stress levels (odds ratio = 1.148; 95% confidence interval = 1.094, 1.205) than those without such experience, and more suicidal thoughts (odds ratio = 1.355; 95% confidence interval = 1.266, 1.450), suicide planning (odds ratio = 1.717; 95% confidence interval = 1.527, 1.929), and suicide attempts (odds ratio = 1.852; 95% confidence interval = 1.595, 2.151). Thus, it is important to pay increased attention to mental health and suicide-related issues in South Korean adolescents with part-time jobs.

## 1. Introduction

Worldwide, 23% of youth in Organization for Economic Cooperation and Development countries are engaged in some form of employment [1]. In the United States, among youth aged 16 to 19, this figure stands at 35.2% [2]. In Korea, about 1.47 million youth are employed, and the share of adolescents in the workforce has been increasing in recent years [3].

There are conflicting views on the impact of youth employment [4]. On the one hand, it can be an opportunity to learn life lessons, such as those related to accountability, financial literacy, and time management [5]. Indeed, steady work for low-income adolescents is associated with a more positive view of the future and higher efficacy [6]. On the other hand, youth employment has been associated with low academic achievement, misconduct, and suicidal thoughts [7,8,9]. In an American study, adolescents working more than 20 h a week had lower academic performance than those who worked fewer hours or those who did not work [7]. In Korea, adolescent employment has been shown to be significantly associated with high drinking and smoking rates [9].

Employment patterns and working conditions in adults are significantly associated with mental health and suicidal behavior [10,11]. Work-related stresses such as physically detrimental working conditions, high work tensions [12], and low levels of employment protection [13] are harmful for mental health [10] and increase suicide rates and mortality [14,15]. Especially, it is common for adolescents to be engaged in low-wage work in services and retail, where the environment is difficult to monitor [8]. Additionally, the employment of adolescents is generally kept under wraps and is characterized by a high degree of dependence on employers; thus, there is a risk of violence exposure [16]. In fact, in a study of South Korean youth, students with part-time jobs had a higher risk of violence exposure than those without part-time jobs [17]. Violence exposure in adolescents is not restricted to physical trauma; it extends to mental health impairment and increased risk of suicidal behavior [18,19,20].

The level of employment protection for South Korean youth can generally be regarded as poor [8]. According to the Children and Adolescents Human Rights Survey (2019), 14.6% of adolescents received either no wages or less than what was promised; 12.4% were forced to work overtime, which was in violation of their contract; 10.4% were exposed to a dangerous or unclean work environment; and 10.2% experienced verbal abuse. Taken together, South Korean adolescents are exposed to poor working conditions without access to adequate protective services [21].

Suicide is the leading cause of adolescent death, with a high mortality rate of 22.8 per 100,000 people [22]. In 2018, the suicide rate in Korean adolescents was 13.3%, while the rate of attempted suicide was 3.1% [23]. In particular, working adolescents are vulnerable to mental health problems and suicide because they not only experience high levels of physical and mental stress but also often work in unprotected environment. Indeed, previous studies have reported that work experience increases suicide attempts in middle school students [8]. However, results regarding the impact of part-time work in adolescents remain contradictory. According to Monahan et al., the reported inconsistencies stem from selection bias [24].

Previous studies on the associations between part-time work, mental health problems, and suicidal behavior in adolescents have mostly employed nonrandom sampling, which involves the aforementioned problem of selection bias [8,10,12,13]. Thus, there is the possibility of incorrect inferences about causality or errors leading to the underestimation or overestimation of results. Propensity score matching (PSM) is a method of balancing covariates between the experimental and control groups in a real-life situation where a perfect, randomized, experimental design is not possible. This method has an advantage in that it enables the determination of the influence of the independent variable by minimizing selection bias, which may affect the results. Therefore, in this study, the basic demographic differences between adolescents with and without part-time work experience were revised using PSM. Then, the aim was to determine how part-time work experience is associated with mental health and suicidal behavior in adolescents.

## 2. Materials and Methods 

### 2.1. Design and Population

This study used data from the 11th–13th Korea Youth Risk Behavior Web-Based Surveys (KYRBWS-XI–XIII) performed by the Korea Centers for Disease Control and Prevention between 2015 and 2017. The KYRBWS, initiated in 2005, is an annual, anonymous, online survey to evaluate the health behaviors of South Korean adolescents, from middle school sophomore to high school senior. This nationwide cross-sectional survey is conducted with students from 400 middle schools and 400 high schools. In 2015, approximately 68,000 middle and high school students from 797 sample schools participated. The survey consisted of approximately 120 questions in 15 areas including smoking, drinking, physical activity, eating, and mental health.

### 2.2. Measures

#### 2.2.1. Part-Time Work 

Part-time work experience was determined using one question: “Have you had a part-time job in the past year?” The possible answers were “yes” or “no.” 

#### 2.2.2. Mental Health 

Since the secondary data used in this study did not cover each individual’s medical/psychological diagnosis, “self-reported risk behavior” and “psychological distress” were used to assess mental health. Self-reports of high-risk behaviors, defined as deviant or problematic behaviors [25], included lifetime drinking, problem drinking, lifetime smoking, sexual intercourse experience, and violence exposure. The questions about lifetime drinking, lifetime smoking, and sexual intercourse experience were “Have you ever drank/smoked/had sexual intercourse?” The response options were “yes” or “no.” Binge drinking was evaluated by the question, “What was your average alcohol consumption in the last month?” The response options ranged from “1–2 glasses” to “˃12 glasses.” In this study, based on the recommendation of the National Institute on Alcohol Abuse and Alcoholism (2004), consumption of more than five glasses of alcohol on a single occasion was considered binge drinking. Violence exposure was determined by the question, “Have you ever been treated in a hospital for physical violence, intimidation, bullying, and so on by a friend, senior, or adult in the past 12 months? If yes, how many times have you been treated?” Possible answers ranged from “0” to “˃6 times.” Thus, in this study, the data did not include violence exposure without hospital treatment.

The variable of psychological pain commonly encompasses stress, depression, and anxiety [26]. In this study, psychological distress included subjective health status, subjective happiness status, stress awareness, depression awareness, and sleep time satisfaction. Subjective health status and subjective happiness status were categorized into “good” and “bad.” Stress awareness was categorized as “high” and “low.” Depression awareness was measured by the question “Have you ever felt sad or desperate enough to stop your daily life for two consecutive weeks in the past year?” The response options were “yes” or “no.” Sleep satisfaction was classified as “yes” or “no.”

#### 2.2.3. Suicidal Behavior 

Suicidal behavior was investigated through three domains: Suicidal ideation, suicide planning, and suicide attempts. Suicidal ideation was measured by the question, “Have you ever seriously considered suicide during the last year?” Suicide planning was investigated by the question “Have you ever made any specific plans for suicide in the past year?” Finally, suicide attempts were assessed by the question, “Have you ever attempted suicide during the last 12 months?” The response options for all questions were “yes” or “no.” Therefore, the data used to measure suicidal behavior in this study did not cover method, severity, or frequency. 

### 2.3. Covariates

Age was categorized into six groups (13 to 18 years old). School type was categorized into three groups (middle school, high school, and vocational high school). Living arrangement was categorized into four groups (with both parents, with a single parent, with grandparent(s) or other relatives, and other, such as living alone). Perceived socioeconomic status and school performance were categorized into three groups (high, middle, and low). Parental educational level was classified into three groups: Below middle school, high school, and above college. Weekly allowance, determined by the question, “How much is the average weekly allowance that you can use freely, excluding transportation and cell phone charges?” was categorized into three groups (under 50$, 50$–100$, and ˃100$).

### 2.4. Data Analysis

All statistical analyses were conducted with SPSS version 26.0 (IBM Corp., Armonk, NY, USA).

#### 2.4.1. PSM 

This statistical method, developed by Rosenbaum and Rubin in 1983, is based on the assumption that when an individual belongs to a group, he or she may have a basic tendency to belong to that group. PSM minimizes selection bias by matching tendencies that may affect the experimental and control groups. In other words, in analyzing a realistic situation in which a fully controlled experimental design is impossible, this method is advantageous in that the influence of the independent variable can be more clearly identified by removing a particular problem that may affect the result. 

In this study, first, self-reported risk behavior (violence exposure, lifetime drinking, problem drinking, lifetime smoking, sexual intercourse experience, subjective happiness status, subjective health status, stress awareness, depression awareness, and sleep time satisfaction) and psychological distress (suicidal ideation, suicide planning, and suicide attempts) were used as the dependent variables, and sociodemographic variables (age, school type, residential area, living arrangement, perceived socioeconomic status, school performance, father’s educational level, mother’s educational level, and weekly allowance) were used as covariates in PSM. Second, treatment case (part-time job experience) was matched with control case (no part-time job experience) by using 1:1 nearest neighbor matching. Of the 141,967 samples, 34,330 were reconstructed through this matching process. Finally, the results of logistic regression analysis between the experimental and control groups for mental health and suicidal behavior were compared. 

#### 2.4.2. The χ2 Test 

The χ2 test was performed to compare the covariate distribution before and after PSM.

#### 2.4.3. Multiple Logistic Regression 

Multiple logistic regression was performed to analyze the associations between part-time work, mental health, and suicidal behavior among Korean adolescents after PSM. The independent variable was part-time work and the dependent variables were mental health and suicidal behavior. The covariates were sociodemographic factors such as age, school type, residential area, living arrangement, perceived socioeconomic status, academic performance, father’s educational status, mother’s educational status, and weekly allowance. First, a logistic regression model was built using the full (pre-matching) sample of Korean adolescents (from middle school freshmen to high school graduates). Second, after PSM, a logistic regression analysis was performed by using a reduced (post-matching) sample containing only those cases included in the matches.

### 2.5. Ethical Consideration

This study was exempted of the institutional review board from author’s university (IRB No. 1041078-202006-HR-159-01)

## 3. Results

Table 1 shows the demographic characteristics of adolescents with and without part-time job experience. Of the 17,165 students with part-time experience, 86.6% were high school students, and as they got older, part-time work experience also increased. In the group with part-time job experience, 14.9% lived with one parent and 2.0% lived with grandparents or relatives. This is higher than the 7.8% who lived with one parent and 0.9% who lived with grandparents or relatives in the group without part-time job experience. The percentage of respondents who perceived their socioeconomic status as low was 23.2% in the group with part-time job experience, which was higher than the 11.5% in the group without part-time job experience. Even in the case of academic achievement, 46.4% of those with part-time job experience reported low levels, which was considerably higher than the 26.8% in the group without part-time job experience. Also, at 3.9%, the percentage of respondents with part-time work experience who answered “yes” to violence exposure experience was more than double the 1.7% in the group without part-time work experience. Overall, there was a statistically significant difference in general characteristics between the groups with and without part-time job experience.

Table 2 shows the results of the χ2 test of general characteristics according to part-time job experience before and after matching to confirm that the PSM was performed well. There were 141,967 samples before matching; subsequently, 34,330 remained. Pre-matching, 13.0% of males and 11.4% of females reported having a part-time job. The χ2 test revealed a significant difference in gender, age, school type, residential area, living arrangement, perceived socioeconomic status, academic achievement, father's educational level, mother's educational level, and weekly allowance before matching according to part-time job experience. Later, there was no statistically significant difference except with regard to school type, living arrangement, and father’s educational level. This is because when there are large variations in group characteristics, a certain difference may occur even when matching according to the propensity score. Compared with the pre-matching sample, differences in covariates between the groups with and without part-time job experience decreased, as shown by the lower ratio of variables and larger p-value for the χ2 test in the post-matching sample.

Table 3 shows the difference in mental health and suicidal behavior between the groups with and without part-time job experience through logistic regression after homogenizing their general characteristics through PSM. Overall, part-time experience had a significant negative relationship with mental health variables and suicidal behavior. Subjects with part-time job experience were more likely to have high subjective health status, but their sleep satisfaction and subjective happiness were lower than the other group. Further, they had 1.148 (1.094, 1.205 95% confidence interval (CI)) times higher stress awareness and 1.544 (1.463, 1.630) times higher depression awareness than the group without part-time job experience. In particular, the lifetime drinking rate and problem drinking rate were more than 3.070 (2.912, 3.236) and 3.818 (3.551, 4.106) times higher, respectively, and the smoking rate was also 3.658 (3.447, 3.883) times higher in the group with part-time job experience.

Table 3 also describes the link between part-time job experience and suicidal behavior. Compared to the other group, the group with part-time job experience had a 1.355 (1.266, 1.450) times higher probability of having suicidal ideation, a 1.717 (1.527, 1.929) times higher probability of suicide planning, and 1.852 (1.595, 2.151) times higher suicide attempt rates.

## 4. Discussion

Using data from the KYRBWS-XI-XIII, this cross-sectional survey was conducted to investigate the association between part-time work experience, mental health, and suicidal behavior in adolescents. According to the results of PSM, part-time work experience had a positive association with poor mental health and suicidal behavior.

Part-time work experience was significantly associated with mental health-related factors such as violence exposure, drinking, smoking, and sexual intercourse experience. Subjects with part-time work experience had a 2.2 times higher probability of having experienced victimization than those without. Considering the National Youth Policy Institution’s (2019) finding that 10.2% of adolescents had experienced verbal abuse and personal insults, these results suggest that adolescents may be exposed to hierarchical violence by employers and clients in the workplace [27]. This corresponds to the findings of a previous study on adolescents wherein part-time employment was positively associated with violence exposure and negatively associated with stress awareness, depression awareness, and suicidality [17]. Also, adolescents with part-time job experience had more than three times higher rates of drinking experience, binge drinking, smoking experience, and sexual experience than those without part-time work experience. These results are consistent with Lee’s finding [9] that middle school students with part-time jobs have higher drinking and smoking rates. In general, the relationship between adolescents’ part-time work experience and delinquency can be explained by the theory of differential socialization [9,28]. According to this theory, any deviant behavior is learned through interaction with others, especially through a communication process within an intimate personal group [29]. Given the general characteristics of adolescents with part-time work experience, such as lower academic achievement and a higher percentage of vocational school admissions (Table 1), they may have relatively more frequent interactions with less academically motivated colleagues. Within this intimate group, adolescents may learn socially undesirable behaviors and engage in problem behaviors. 

Mental health factors and suicidal behaviors were also significantly associated with part-time experience. Adolescents with part-time work experience showed relatively low sleep time satisfaction and low subjective happiness status. In addition, they had a relatively high frequency of stress and depression awareness. Especially, adolescents with part-time work experience showed a higher frequency of suicidal behaviors. As mentioned by Jo et al., sociocultural differences across countries cannot be excluded when considering the various views on adolescent employment [8]. In Western countries, it has been noted that adolescent employment may have beneficial effects on the developmental process and future employment outcomes, especially in low-income families [6,30]. On the contrary, in South Korea’s sociocultural context, part-time work in this population has been considered negative in most cases. First, part-time work usually refers to short-term hourly jobs regardless of whether they are during the semester or vacation. These jobs generally involve poor working conditions that are not related to the acquisition of skills for future employment. Second, generally, adolescents who work part-time tend to be in the minority because of the social pressure concerning academic achievement. In South Korean society, adolescents who work part-time are considered “losers” in the context of the competitive education system, leading to a negative sociocultural perception [9]. Third, unlike in Western countries such as the United States, social safety for adolescents who work part-time is inadequate [8]. In fact, in a previous study of South Korean adolescents, those with part-time job experience had 2.97 (2.63, 3.36, 95% CI) times higher violence exposure risk than those who did not [17]. As a result, in the workplace, adolescents may have a higher probability of exposure to violence, as well as experiencing stigmatization stemming from their deviation from the generally accepted social norms as “students.” This may weaken their social support system and lead them toward suicidal behavior.

This study is of great significance because of the limited global research on the association between part-time work experience and mental health and suicidal behavior in adolescents. However, the results should be interpreted within the context of some limitations. First, as this was a cross-sectional study, we could not make inferences about the causal relationships between part-time work, mental health problems, and suicidal behavior among South Korean adolescents. Also, because of the limited scope of the questions, it was only possible to confirm whether or not adolescents had part-time work experience; there were no data about intensity, duration, or type of part-time work, as well as workplace conditions and working hours. Therefore, it is necessary to clarify the relationship between mental health and suicidal behavior according to the intensity, environment, and type of part-time job through a follow-up study. In addition, even though violence exposure is a potential risk factor for mental health problems, and adolescents who work part-time have a high risk of exposure to violence [17], we did not account for violence exposure that did not require hospitalization. Therefore, future studies should include a comprehensive question on exposure to violence, with detailed analyses according to the type and degree of violence. Second, the data were self-reported and anonymized; thus, it is possible that some participants may not have provided honest responses about their mental health condition and suicidal behavior. Finally, we did not consider factors such as peers, parenting, or community, which may affect mental health and suicidal behavior. Therefore, further research is needed on more detailed factors that can be related to mental health problems and suicidal behavior in working adolescents, such as stress related to the stigma of being a “dropout” or the burden of balancing academics and employment. In the cultural context, we should reconsider the meaning of academic achievement for the working adolescent, and carefully determine the major sources of stress in this population.

Nevertheless, in this study, we surveyed a nationally representative sample, using PSM to minimize selection bias and further refine the relationship between adolescents’ part-time work experience, mental health, and suicidal behavior. Overall, our results suggest that in adolescents, part-time work experience is associated with suicidal behavior as well as mental health problems. From this point of view, there is a need for not only social and institutional mechanisms to effectively protect working adolescents but also social systems to promote their mental health.

## 5. Conclusions

In Korean adolescents, part-time work experience was significantly associated with mental health problems and suicidal behavior even after correcting the difference between the two groups through PSM. A strong connection was found between part-time work experience, mental health problems, and mental health-related factors. Additionally, adolescents with part-time work experience showed a relatively higher risk of suicidal behavior.

This study confirmed the influence of part-time work experience by minimizing the impact of sociocultural background using PSM and supports the need for social protection for working adolescents. Furthermore, it is necessary to investigate the risk factors associated with part-time work experience that can affect adolescent suicide and to further study interventions that can effectively protect working youth from these risk factors. In addition, since few studies have dealt with adolescents’ part-time work experience, mental health, and suicidal behavior together, we tried to determine the overall tendency of the association of part-time work with mental health and suicidal behavior in this population. As it is commonly known that gender has a significant influence on suicidal behavior [31], it will be interesting to analyze adolescents’ part-time work experience, mental health problems, and suicidal behavior according to gender in future studies.

## Figures and Tables

**Table 1 ijerph-17-06835-t001:** Summary of demographic and socioeconomic characteristics in the groups with and without part-time job experience (*N* = 141,967).

Variable	Classification	Without Part-Time Job Experience (*n* = 124,802)	With Part-Time Job Experience (*n* = 17,165)	χ2	*p*
		% (SE)	% (SE)		
Gender	Male	50.5 (0.8)	54.4 (1.2)	93.336	<0.001
	Female	49.5 (0.8)	45.6 (1.2)		
Age (Years)	13	14.1 (0.2)	1.5 (0.1)	8246.856	<0.001
	14	15.5 (0.2)	3.4 (0.2)		
	15	17.2 (0.2)	8.6 (0.3)		
	16	18.1 (0.2)	19.5 (0.3)		
	17	17.4 (0.2)	32.4 (0.4)		
	18	17.7 (0.2)	34.7 (0.4)		
School Type	Middle School	46.8 (0.5)	13.4 (0.4)	14,144.348	<0.001
	High School	47.6 (0.5)	57.6 (0.8)		
	Vocational High School	5.6 (0.2)	29.0 (0.7)		
Residential Area	Big City	45.0 (0.5)	36.7 (0.7)	683.185	<0.001
	Medium-Sized City	50.0 (0.5)	54.5 (0.9)		
	Rural Area	5.0 (0.3)	8.8 (0.6)		
Living Arrangement	With both Parents	91.3 (0.1)	83.2 (0.3)	118,660.848	<0.001
	With a Single Parent	7.8 (0.1)	14.9 (0.3)		
	With Grandparent(s)/Other Relatives	0.9 (0.1)	2.0 (0.1)		
	Other				
Perceived Socioeconomic Status	High	43.3 (0.3)	29.6 (0.4)	2297.412	<0.001
	Middle	45.2 (0.2)	47.2 (0.4)		
	Low	11.5 (0.1)	23.2 (0.4)		
Academic Performance	High	44.5 (0.2)	26.5 (0.4)	3183.205	<0.001
	Middle	28.8 (0.1)	27.2 (0.3)		
	Low	26.8 (0.1)	46.4 (0.4)		
Father’s Educational Level	Below Middle School	2.1 (0.0)	4.8 (0.2)	3018.126	<0.001
	High School	30.7 (0.3)	48.9 (0.5)		
	Above College	67.2 (0.3)	46.3 (0.5)		
Mother’s Educational Level	Below Middle School	1.8 (0.0)	4.0 (0.2)	2548.592	<0.001
	High School	38.7 (0.3)	56.2 (0.4)		
	Above College	59.5 (0.3)	39.8 (0.5)		
Weekly Allowance	<50$	86.7 (0.1)	72.4 (0.4)	2679.599	<0.001
	50$–100$	9.6 (0.1)	17.4 (0.3)		
	≥100$	3.7 (0.1)	10.2 (0.2)		

SE: Standard error.

**Table 2 ijerph-17-06835-t002:** Covariate imbalance before and after propensity score matching.

	Classification	Pre-Matching Sample (*n* = 141,967)			Post-Matching Sample (*n* = 34,330)		
Covariates: % (SE)		With Part-Time Job Experience	Without Part-Time Job Experience	*p Value*	With Part-Time Job Experience	Without Part-Time Job Experience	*p Value*
Gender	Male	13.0 (0.3)	87.0 (0.3)	<0.001	49.6 (0.6)	50.4 (0.6)	0.955
	Female	11.4 (0.3)	88.6 (0.3)		49.6 (0.7)	50.4 (0.7)	
Age (Years)	13	1.4 (0.1)	98.6 (0.1)	<0.001	50.1 (2.3)	49.9 (2.3)	0.001
	14	3.0 (0.1)	97.0 (0.1)		48.9 (1.6)	51.1 (1.6)	
	15	6.5 (0.2)	93.5 (0.2)		50.7 (1.1)	49.3 (1.1)	
	16	13.0 (0.3)	87.0 (0.3)		46.8 (0.7)	53.2 (0.7)	
	17	20.6 (0.4)	79.4 (0.4)		50.4 (0.7)	49.6 (0.7)	
	18	21.4 (0.4)	78.6 (0.4)		50.3 (0.7)	49.7 (0.7)	
School Type	Middle School	3.8 (0.1)	96.2 (0.1)	<0.001	50.2 (0.9)	49.8 (0.9)	<0.001
	High School	14.4 (0.3)	85.6 (0.3)		48.3 (0.6)	51.7 (0.6)	
	Vocational High School	42.0 (0.9)	58.0 (0.9)		52.1 (0.9)	47.9 (0.9)	
Residential Area	Big City	10.2 (0.2)	89.8 (0.2)	<0.001	50.4 (0.7)	49.6 (0.7)	0.384
	Medium-Sized City	13.2 (0.2)	86.8 (0.3)		49.2 (0.6)	50.8 (0.6)	
	Rural Area	19.5 (1.0)	80.5 (1.0)		48.5 (1.6)	51.5 (1.6)	
Living Arrangement	With Both Parents	11.2 (0.2)	88.2 (0.2)	<0.001	49.2 (0.5)	50.8 (0.5)	0.003
	With a Single Parent	21.0 (0.4)	79.0 (0.4)		52.0 (0.8)	48.0 (0.8)	
	With Grandparent(s)/Other Relatives	23.6 (1.4)	76.4 (1.4)		47.3 (2.2)	52.7 (2.2)	
Perceived Socioeconomic Status	High	8.7 (0.2)	91.3 (0.2)	<0.001	50.3 (0.6)	49.7 (0.6)	0.088
	Middle	12.7 (0.2)	87.3 (0.2)		48.9 (0.6)	51.1 (0.6)	
	Low	21.9 (0.4)	78.1 (0.4)		50.1 (0.7)	49.9 (0.7)	
School Performance	High	7.6 (0.2)	92.4 (0.2)	<0.001	50.7 (0.7)	49.3 (0.7)	0.051
	Middle	11.6 (0.2)	88.4 (0.2)		49.6 (0.7)	50.4 (0.7)	
	Low	19.4 (0.3)	80.6 (0.3)		48.9 (0.5)	51.1 (0.5)	
Father’s educational level	Below middle School	24.3 (0.8)	75.7 (0.8)	<0.001	45.9 (1.4)	54.1 (1.4)	0.023
	High School	18.1 (0.3)	81.9 (0.3)		49.7 (0.6)	50.3 (0.6)	
	Above College	8.8 (0.1)	91.2 (0.1)		49.8 (0.6)	50.2 (0.6)	
Mother’s Educational Level	Below Middle School	23.7 (0.8)	76.3 (0.8)	<0.001	46.4 (1.5)	53.6 (1.5)	0.071
	High School	16.8 (0.3)	83.2 (0.3)		49.5 (0.5)	50.5 (0.5)	
	Above College	8.5 (0.2)	91.5 (0.2)		50.0 (0.6)	50.0 (0.6)	
Weekly Allowance	<50$	10.4 (0.2)	89.6 (0.2)	<0.001	49.5 (0.5)	50.5 (0.5)	0.434
	50$–100$	20.1 (0.4)	79.9 (0.4)		49.3 (0.8)	50.7 (0.8)	
	>100$	27.8 (0.6)	72.2 (0.6)		50.7 (1.0)	49.3 (1.0)	

SE: Standard error.

**Table 3 ijerph-17-06835-t003:** Mental health and suicidal behavior by part-time experience after propensity score matching (*n* = 34,330).

	Adjusted Odds Ratio	95% CI	*p*
Violence Exposure	2.284	1.959, 2.663	<0.001
Lifetime Drinking (%)	3.070	2.912, 3.236	<0.001
Problem Drinking (%)	3.818	3.551, 4.106	<0.001
Lifetime Smoking (%)	3.658	3.447, 3.883	<0.001
Sexual Intercourse Experience	3.258	2.971, 3.571	<0.001
Sleep Time Satisfaction (%)	0.836	0.786, 0.889	<0.001
Subjective Happiness Status (%)	0.909	0.865, 0.955	<0.001
Subjective Health Status (%)	1.110	1.053, 1.169	<0.001
Stress Awareness (%)	1.148	1.094, 1.205	<0.001
Depression Awareness (%)	1.544	1.463, 1.630	<0.001
Suicidal Ideation	1.355	1.266, 1.450	<0.001
Suicidal Planning	1.717	1.527, 1.929	<0.001
Suicide Attempts	1.852	1.595, 2.151	<0.001

CI: Confidence interval.

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
