# Peer review of "Adolescent Employment, Mental Health, and Suicidal Behavior: A Propensity Score Matching Approach"

_ijerph, 2020, doi:10.3390/ijerph17186835_

Round 1
Reviewer 1 Report
The article "Adolescent Employment, Mental Health, and Suicidal Behavior: A Propensity Score-Matching Approach" deals with a topic of great relevance at present, such as mental health problems and suicidal behavior in adolescents in relation, in this case, to the part-time work experience.
Then, I propose some suggestions that could improve the manuscript presented:
- In the last paragraph of the introduction it would be interesting to note again the studies referred to for ease of reading.
- Part-Time Work: it would be interesting to have assessed how long you were working on part-time work experience.
- Justify this statement in the study "the data did not include victimization experience without hospital treatment". It seems relevant that, even if there was no hospitalization, it can be considered a victimization experience. Justify with some study why it has not been considered.
- Mental health questions are very subjective, perhaps for future studies we recommend using tools validated for this type of adolescent population such as the BDI (Beck), STAI, Paykel (PSS, Paykel) or other similar questionnaires.
- I recommend including in each section why those items have been selected. Previous studies, scientific literature supporting these issues, etc.
- Has socio-economic status been measured only subjectively? Or, depending on income, was it in one position or another as has been done with the weekly allowance?
- It might be interesting to analyze the results in terms of gender. Are there any previous studies that consider these differences in terms of gender? Or, if they have already been analyzed and no relevant differences are found, point this out throughout the text to justify not doing so in this study.
- It would be interesting to know whether the respondents were already suffering from these symptoms before they started this job or whether they can be said to be a "consequence of".
Author Response
[September 14, 2020]
Prof. Dr. Paul B. Tchounwou
Editor-in-Chief
International Journal of Environmental Research and Public Health
Dear Editor-in-Chief:
Thank you for the opportunity to revise and resubmit our manuscript entitled “Adolescent Employment, Mental Health, and Suicidal Behavior: A Propensity Score Matching Approach” for consideration as a research article (ijerph-919692).
We thank you and the reviewers for your suggestions and insights, which have enriched the manuscript and produced a more balanced and stronger account of our research. We hope that the revised manuscript is now suitable for publication in International Journal of Environmental Research and Public Health.
The manuscript has been carefully checked and the necessary changes have been made in accordance with the reviewers’ suggestions. The responses to all comments have been prepared and attached herewith.
Thank you for your consideration. We look forward to hearing from you soon.
Sincerely,
Hyerine Shin, RN, MS
Department of Nursing, Graduate School, Chung-Ang University
84 Heuksuk-ro Dongjak-gu, 06974 Seoul, South Korea
M: +82-10-7117-5088, E: [email protected]
Reviewer #1
Thank you for your extensive review and constructive comments, which have enhanced the clarity and organization of our manuscript.
- In the last paragraph of the introduction it would be interesting to note again the studies referred to for ease of reading.
Response: We have revised the Introduction to include the relevant references in the final paragraph.
Page 2, lines 66–68: Previous studies on the associations between part-time work, mental health, and suicidal behavior in adolescents have mostly employed non-random sampling, which involves the aforementioned problem of selection bias [8,10,12,13].
- Part-Time Work: it would be interesting to have assessed how long you were working on part-time work experience.
Response: While we agree that the inclusion of the duration of part-time work experience would have been interesting, we were unable to use this variable as this study involved a secondary analysis of data collected for the 11th–13th Korea Youth Risk Behavior Web-Based Survey; therefore, there were limitations on the aspects we could explore. We have clarified this in the limitations.
Page 8, lines 270–271: …there were no data about intensity, duration, or type of part-time work, as well as workplace conditions and working hours.
- Justify this statement in the study "the data did not include victimization experience without hospital treatment". It seems relevant that, even if there was no hospitalization, it can be considered a victimization experience. Justify with some study why it has not been considered.
Response: We agree that victimization experience (violence exposure) without hospitalization should be considered. However, as ours was a secondary analysis of raw data collected for the purpose of the 11th–13th Korea Youth Risk Behavior Web-Based Survey, there were variables that, although of interest to this study, were not available for analysis because they were not included in the original dataset. This has been described under limitations.
Page 9, lines 273–278: In addition, even though violence exposure is a potential risk factor for mental health problems, and adolescents who work part-time have a high risk of exposure to violence [17], we did not account for violence exposure that did not require hospitalization. Therefore, future studies should include a comprehensive question on exposure to violence, with detailed analyses according to the type and degree of violence.
- Mental health questions are very subjective, perhaps for future studies we recommend using tools validated for this type of adolescent population such as the BDI (Beck), STAI, Paykel (PSS, Paykel) or other similar questionnaires.
Response: Our study has certain limitations owing to its reliance on secondary data from the 11th–13th Korea Youth Risk Behavior Web-Based Survey. As the Korea Youth Risk Behavior Web-Based Survey is a large-scale study conducted with the aim of examining various aspects of youth health, the use of the tools mentioned, which are specific to certain conditions, may not have been possible. It is hoped that in the future, research using more sophisticated mental health measurement tools will be able to confirm objective rather than subjective mental health levels.
- I recommend including in each section why those items have been selected. Previous studies, scientific literature supporting these issues, etc.
Response: We have added the rationale for item selection.
Page 3, lines 92–96: Since the secondary data used in this study did not cover each individual’s medical/psychological diagnosis, “self-reported risk behavior” and “psychological distress” were used to assess mental health. Self-reports of high-risk behaviors, defined as deviant or problematic behaviors [25], included lifetime drinking, problem drinking, lifetime smoking, sexual intercourse experience, and violence exposure.
Page 3, lines 107–109: The variable of psychological pain commonly encompasses stress, depression, and anxiety [26]. In this study, psychological distress included subjective health status, subjective happiness status, stress awareness, depression awareness, and sleep time satisfaction.
- Has socio-economic status been measured only subjectively? Or, depending on income, was it in one position or another as has been done with the weekly allowance?
Response: Socioeconomic status was measured only subjectively. Owing to the aforementioned limitation posed by the use of secondary data, we were unable to obtain any further details, such as car ownership or family property ownership, to confirm objective socioeconomic position. However, as many studies have reported an association between mental health and subjective economic status and as subjective socioeconomic levels can be more important to adolescents, who may not be able to obtain objective information about their families’ socioeconomic status, the non-use of objective data is unlikely to have had a significant effect on the results.
- It might be interesting to analyze the results in terms of gender. Are there any previous studies that consider these differences in terms of gender? Or, if they have already been analyzed and no relevant differences are found, point this out throughout the text to justify not doing so in this study.
Response: Since there have been few studies that have dealt with adolescents’ part-time job experience, mental health, and suicidal behavior together, we tried to determine the overall tendency of the association of part-time jobs with adolescents’ mental health problems and suicidal behavior. However, we strongly agree with your opinion that analyzing the results by gender would be meaningful, and we will explore this aspect in future research.
Page 9, lines 303-308: In addition, since few studies have dealt with adolescents’ part-time job experience, mental health, and suicidal behavior together, we tried to determine the overall tendency of the association of part-time work with mental health problems and suicidal behavior in this population. As it is commonly known that gender has a significant influence on suicidal behavior [29], it will be interesting to analyze adolescents’ part-time job experience, mental health problems, and suicidal behavior according to gender in future studies.
- It would be interesting to know whether the respondents were already suffering from these symptoms before they started this job or whether they can be said to be a "consequence of".
Response: Unfortunately, owing to the use of cross-sectional data, we were unable to determine causality. This has been listed as a limitation of the study.
Page 8, lines 266–268: First, as this was a cross-sectional study, we could not make inferences about the causal relationships between part-time work, mental health problems, and suicidal behavior among South Korean adolescents.
- Additionally, we have improved English language and style with Editage.

Reviewer 2 Report
This is a well-written paper on an important topic. The authors clearly explain the methodology and results. I have two primary recommendations for a revision:
- Improve/re-label the indicators of mental health among the respondents. The authors lump together "risky behaviors" (such as alcohol use, smoking and sexual activity) with both risk FACTORS like experience of victimization and self-reports of markers of psychological distress (poor sleep quality, low happiness ranking, etc.) Either provide a justification for lumping all of these together (an accepted index of indicators of poor mental health, for example) or separate them into different categories, such as "self reports of high-risk behaviors" and "self-reports of poor mental health." I recommend the term "psychological distress" since the authors are not actually diagnosing mental health disorders among respondents. The authors should be sure to clarify that they are using these factors as proxy for mental health problems. They should also revise this sentence in the abstract; "Overall, part-time experience was significantly related to mental health and suicidal behavior even after removing background differences." What they mean is that part-time work experience was associated with indicators of poor mental health or mental health problems, not with mental health itself.
- The discussion and conclusion need to be expanded, both to include a more substantial implications section and to address the possible reasons for the findings. For example, is the only recommendation from this study that there should be more workplace protections for adolescent workers? What about the issue related to the stigma of working as a student- the labeling of adolescent workers as "losers?" Could it be the stigma itself that is the cause of the distress and suicidal ideation? Or the stress of managing both a significant academic load and work? Could the distress come from low social class rather than the work or work environment directly? The authors should be clear about the limits of the conclusions that can be drawn from this study but also suggest further research that could actually draw conclusions about causation.
Author Response
[September 14, 2020]
Prof. Dr. Paul B. Tchounwou
Editor-in-Chief
International Journal of Environmental Research and Public Health
Dear Editor-in-Chief:
Thank you for the opportunity to revise and resubmit our manuscript entitled “Adolescent Employment, Mental Health, and Suicidal Behavior: A Propensity Score Matching Approach” for consideration as a research article (ijerph-919692).
We thank you and the reviewers for your suggestions and insights, which have enriched the manuscript and produced a more balanced and stronger account of our research. We hope that the revised manuscript is now suitable for publication in International Journal of Environmental Research and Public Health.
The manuscript has been carefully checked and the necessary changes have been made in accordance with the reviewers’ suggestions. The responses to all comments have been prepared and attached herewith.
Thank you for your consideration. We look forward to hearing from you soon.
Sincerely,
Hyerine Shin, RN, MS
Department of Nursing, Graduate School, Chung-Ang University
84 Heuksuk-ro Dongjak-gu, 06974 Seoul, South Korea
M: +82-10-7117-5088, E: [email protected]
Reviewer #2
Thank you for your extensive review and constructive comments, which have enhanced the clarity and organization of our manuscript.
- Improve/re-label the indicators of mental health among the respondents. The authors lump together "risky behaviors" (such as alcohol use, smoking and sexual activity) with both risk FACTORS like experience of victimization and self-reports of markers of psychological distress (poor sleep quality, low happiness ranking, etc.) Either provide a justification for lumping all of these together (an accepted index of indicators of poor mental health, for example) or separate them into different categories, such as "self-reports of high-risk behaviors" and "self-reports of poor mental health." I recommend the term "psychological distress" since the authors are not actually diagnosing mental health disorders among respondents. The authors should be sure to clarify that they are using these factors as proxy for mental health problems. They should also revise this sentence in the abstract; "Overall, part-time experience was significantly related to mental health and suicidal behavior even after removing background differences." What they mean is that part-time work experience was associated with indicators of poor mental health or mental health problems, not with mental health itself.
Response: We greatly appreciate your detailed comments. As recommended, the mental health variables have been separated into “self-reported risk behavior” and “psychological distress.” Also, for clarity, we have changed “victimization experience” to “violence exposure.” In particular, we have sought to clarify that we are speaking of markers of poor mental health, not a medical diagnosis regarding mental health.
Page 3, lines 92–96: Since the secondary data used in this study did not cover each individual’s medical/psychological diagnosis, “self-reported risk behavior” and “psychological distress” were used to assess mental health. Self-reports of high-risk behaviors, defined as deviant or problematic behaviors [25], included lifetime drinking, problem drinking, lifetime smoking, sexual intercourse experience, and violence exposure.
Page 3, lines 107–109: The variable of psychological pain commonly encompasses stress, depression, and anxiety [26]. In this study, psychological distress included subjective health status, subjective happiness status, stress awareness, depression awareness, and sleep time satisfaction.
Page 4, lines 142–148: In this study, first, self-reported risk behavior (violence exposure, lifetime drinking, problem drinking, lifetime smoking, sexual intercourse experience, subjective happiness status, subjective health status, stress awareness, depression awareness, and sleep time satisfaction) and psychological distress (suicidal ideation, suicide planning, and suicide attempts) were used as the dependent variables, and sociodemographic variables (age, school type, residential area, living arrangement, perceived socioeconomic status, school performance, father’s educational level, mother’s educational level, and weekly allowance) were used as covariates in PSM.
Page 1, lines 17–18: Overall, part-time experience was significantly related to mental health problems and suicidal behavior even after eliminating background differences.
- The discussion and conclusion need to be expanded, both to include a more substantial implications section and to address the possible reasons for the findings. For example, is the only recommendation from this study that there should be more workplace protections for adolescent workers? What about the issue related to the stigma of working as a student- the labeling of adolescent workers as "losers?" Could it be the stigma itself that is the cause of the distress and suicidal ideation? Or the stress of managing both a significant academic load and work? Could the distress come from low social class rather than the work or work environment directly? The authors should be clear about the limits of the conclusions that can be drawn from this study but also suggest further research that could actually draw conclusions about causation.
Response: We have added more limitations of this study, as well as suggestions for further research, under Discussion.
Page 9, lines 273–278: In addition, even though violence exposure is a potential risk factor for mental health problems, and adolescents who work part-time have a high risk of exposure to violence [17], we did not account for violence exposure that did not require hospitalization. Therefore, future studies should include a comprehensive question on exposure to violence, with detailed analyses according to the type and degree of violence.
Page 9, lines 281–286: Therefore, further research is needed on more detailed factors that can be related to mental health problems and suicidal behavior in working adolescents, such as stress related to the stigma of being a “dropout” or the burden of balancing academics and employment. In the cultural context, we should reconsider the meaning of academic achievement for the working adolescent, and carefully determine the major sources of stress in this population.
- Additionally, we have improved English language and style with Editage.
